# Associations between Household Food Insecurity and Depressive Symptomology among Adolescent Girls and Young Women during the COVID-19 Pandemic in South Africa

**Stanley Carries** [1,*], **Lovemore Nyasha Sigwadhi** [2], **Audrey Moyo** [2], **Colleen Wagner** [3], **Catherine Mathews** [1] **and Darshini Govindasamy** [1]

1. Health Systems Research Unit, South African Medical Research Council (SAMRC), Tygerberg 7505, South Africa; catherine.mathews@mrc.ac.za (C.M.); darshini.govindasamy@mrc.ac.za (D.G.)
2. Division of Epidemiology and Biostatistics, Department of Global Health, Faculty of Medicine and Health Sciences, Stellenbosch University, Cape Town 7505, South Africa; lsigwadhi@gmail.com (L.N.S.); 27689387@sun.ac.za (A.M.)
3. Networking HIV and AIDS Community of Southern Africa (NACOSA), Cape Town 7441, South Africa; colleen@nacosa.org.za
* Correspondence: stanley.carries@mrc.ac.za

**Abstract:** Evidence suggests an association between food insecurity and depressive symptomatology; however, little is known about the association between adolescent girls and young women (AGYW) in the context of COVID-19. This study aimed to investigate the relationship between household food insecurity (HFI) and depressive symptomology among AGYW in South Africa during the COVID-19 pandemic. Secondary data analysis was conducted using cross-sectional data collected from the HERStory2 study conducted during the COVID-19 pandemic. The data were collected from 515 AGYW (aged 15–24 years) recruited from six South African districts using a demographic detail and socio-economic questionnaire as well as the Center for Epidemiological Studies Depression (CESD-10) Scale. Data were fitted using a multi-variable robust Poisson regression model and controlled for sociodemographic and health factors. The results suggest that the majority of the sample of AGYW were 20 years old, with 74% exposed to HFI and 30.29% experiencing depressive symptomology. AGYW exposed to HFI were 1.80 times at risk of depressive symptomology compared to those from food-secure households [adjusted risk ratio (aRR): 1.80; 95% CI: 1.35–2.42, $p < 0.0001$)]. Future pandemic-preparedness strategies should incorporate screening for HFI as a means to identify AGYW who may require psychosocial support.

**Keywords:** mental health; household food insecurity; adolescent girls and young women; COVID-19; CES-D-10; depression

## 1. Introduction

The global prevalence of clinical depression among young people (≤18 years) during COVID-19 (25.2%) [1] almost doubled the pre-COVID-19 estimate of 12.9% [2]. COVID-19-related lockdowns disrupted daily routines, restricted social interactions, and limited access to health and social services [3]. The early stages of lockdown were associated with languishing mental health [4], with younger people experiencing higher levels of depression, anxiety, and stress [5]. Adolescents and younger people generally felt more worried and helpless and experienced increased depressive symptomology compared to pre-COVID-19 levels [6].

Globally, mental health disorders affect one in seven adolescents yet remain largely undetected and untreated despite the availability of effective treatments [2,7,8]. Depression is a major contributor to the overall burden of disease and a leading cause of disability worldwide [9,10]. Left untreated during early adolescence, depression can have long-term negative health, economic, and social consequences in adulthood [2]. Economic

consequences include health system costs to treat depression and related comorbidities and reduced contribution to national economic output due to decreased productivity [11]. The social impacts include relationship problems, social isolation, failure to complete school, unemployment, substance misuse, early pregnancy, and substance misuse [12].

Intervening to promote positive mental health during adolescence requires a solid understanding of the correlates of common mental health disorders to develop tailored strategies. Household food insecurity is a known risk factor for mental health disorders [12,13], particularly during adolescence [14] and young adulthood [15]. Research indicates that HFI during the COVID-19 pandemic was associated with increased vulnerability to mental health conditions [16–18]. In the United States, for instance, adults exposed to HFI had 3.53 greater odds of being depressed (95% CI: 2.99–4.17) [17]. This was more pronounced in sub-Saharan Africa (SSA), given the region's pre-existing structural inequalities. A study in Tanzania, for example, found that adolescents living in food-insecure households were at 1.8 times higher risk (95% CI: 1.3–2.5) of experiencing suicidal ideation and at a 2.4 times higher risk (95% CI: 1.7–3.3) of attempting suicide [19]. In addition, a Kenyan study showed that adolescents who skipped meals because of COVID-19-related household income losses had 2.5 higher odds (95% CI: 2.0–3.1) of having depressive symptoms [18]. Against this backdrop, AGYW have been neglected in COVID-19 recovery efforts [20].

AGYW are at a stage of life where they build human and social capital essential for their wellbeing in adulthood [21]. Studies in South Africa have shown that AGYW are more vulnerable to depressive symptoms than their male counterparts [8,10]. Depression may also be a precursor to the high prevalence of HIV infection among South African AGYW [22] because of its association with increased alcohol abuse and risky sexual behaviors [23], which are both risk factors for HIV acquisition [22]. Failure to detect depression early in AGYW could impair their mental health in adulthood. Understanding the correlates of depressive symptoms among AGYW in South Africa can help inform brief mental health screening tools to identify those at risk and link them to care.

High income losses (30%) during the COVID-19 lockdown in South Africa amplified HFI [24,25]. Reports of increased depression levels among adolescents and women due to COVID-19 income losses and HFI in South Africa [26,27], *inter alia*, have increased the urgency for technologies to readily detect AGYW at high risk of depression and, by proxy, households possibly in urgent need of economic support. However, few investigators have quantitatively examined the association between HFI and depressive symptoms among AGYW in this setting during COVID-19. Identifying AGYW at risk of developing depressive symptoms could help identify factors to prevent and/or manage anxiety and depression during adulthood and answer international calls to examine the structural drivers of anxiety and depression among youth [28,29], as well as inform mental health-related intervention designs to support the most vulnerable during pandemics. This study aimed to determine the relationship between HFI and depressive symptomatology among AGYW during the COVID-19 pandemic in South Africa.

## 2. Materials and Methods

### 2.1. Study Setting and Population

The data used in this secondary data analysis were drawn from a cross-sectional study, HERStory 2 [30] (the parent study), which was conducted between 01 December 2020 and 28 February 2021 during the COVID-19 lockdown in South Africa. The HERStory 2 study evaluated a combination HIV-prevention intervention (the My Journey programme) for AGYW aged 15–24 years, from six districts spread across six South African provinces: Bonjanala District (NorthWest Province), Klipfontein (Western Cape Province), King Cetshwayo (KwaZulu-Natal Province), Ehlanzeni District (Mpumalanga Province), Nelson Mandela Bay (Eastern Cape Province), and Thabo Mofutsanyana District (Free State Province).

The HERStory 2 study sampling frame was AGYW who accessed the My Journey programme for at least one year and gave permission for telephonic follow-up (n = 2160). In the HERStory 2 study, a team of female interviewers (aged 20–35 years) fluent in

all languages spoken in the sampled districts, conducted telephonic follow-ups with all potentially eligible AGYW. Only those who provided verbal consent were eligible to participate. Of the potentially eligible AGYW, 23.8% (n = 515) were successfully contacted and participated in the study. All interviews were conducted once the AGYW self-reported being in a safe and comfortable place to be administered the telephonic survey. The survey took approximately an hour to complete. A reimbursement amount of ZAR 100 was sent to each participant using electronic banking. For this secondary data analysis, the entire study population in the dataset was used, and no restrictions were placed on age or region.

### 2.2. Measures

The data from the cross-sectional study (HERStory2) were collected with the assistance of study staff. The participants completed (i) a demographic detail and socio-economic questionnaire as well as (ii) the CES-D-10 Scale.

### 2.3. Demographic Detail and Socio-Economic Questionnaire

The demographic detail and socio-economic questionnaire was made up of 271 items, which included the following sub-sections (see Table A1 for the description of variables):

- Demographic variables included age at recruitment, highest school grade completed, enrolled in full-time study before lockdown, disengaged in secondary or tertiary education in 2020, and length of stay in the community.
- Clinical variables included self-reported HIV status, lacked access to essential medication, and lacked access to family planning services.
- Socio-economic variables included lacked access to social support services, violence at home, HFI (which was assessed based on participants' responses to the statement, "During the lockdown I worried that my food would run out because of a lack of money") and household asset index score. "No meals for the whole day" was used as a proxy for HFI in the sensitivity analysis.

### 2.4. CES-D-10 Scale

Depressive symptomology was assessed using the CES-D-10 scale. This scale is a short screening tool consisting of ten items assessing an individual's feelings over the past week. Eight items covered negative feelings (i.e., feeling bothered, lack of focus, feeling depressed, feeling that everything was an effort, feeling fearful, restless sleep, loneliness, and inability to get going), and two items (questions 5 and 8) covered positive feelings (i.e., feeling happy and hopeful about the future). The response options for the scale increased in severity from not/hardly experiencing a depressive symptom (i.e., less than a day) to experiencing the depressive symptom almost all the time (i.e., 5–7 days a week). The CES-D-10 scale was psychometrically evaluated in a South African sample and showed good validity and reliability [31]. A cutoff point of 12 was deemed optimal for detecting the presence of significant depressive symptoms in this sample [31].

## 3. Statistical Analysis

The following variables were dichotomized: *highest school grade completed* ($\geq$Grade 9 = 1 vs. <Grade 9 = 0), *length of stay in community* (<5 years vs. $\geq$5 years), *HIV status* (positive = 1 or negative = 0), *lacked access to social support services* (yes = 1, no = 0), *disengaged in secondary or tertiary education in 2020* (yes = 1 vs. no = 0); each of *lacked access to essential medication, lacked access to family planning services, violence at home, no meal for a whole day*, and *HFI* as (often/sometimes = 1 vs. never = 0). *Age at recruitment* was treated as a continuous variable.

Continuous variables were expressed as medians with interquartile ranges because they were not normally distributed. Categorical variables were expressed as frequencies and percentages. Chi-squared and Fisher's exact tests were used to assess the association between depressive symptoms and categorical variables. The Mann–Whitney U test was used to assess median differences. Owing to the high prevalence of depressive symptoms (30%), logistic regression overestimated the effect measure with large standard errors,

resulting in wide confidence intervals, and log-binomial regression faced convergence problems. Robust Poisson regression was used as an alternative to assess the associations between demographic and socio-economic variables with depressive symptoms. Known factors linked to depressive symptoms, such as socio-economic [32], education, social inclusivity [33], age, and gender [33], were regarded as a priori confounders in the model. Three different models were fitted and adjusted risk ratios with 95% CIs were used as a measure of association. In the first model (Model 1), the following variables were adjusted for *age at recruitment, length of stay in community, highest school grade completed*, and *household asset index score*. In Model 2, *lacked access to social support services* was added to Model 1. Finally, in Model 3, *lacked access to essential medication* was added to Model 2. Multicollinearity was assessed using variance inflation factors (VIF), with a VIF of less than five considered adequate for suggesting no multicollinearity [34]. All statistical analyses were performed using STATA software (version 16, Stata Corp., College Station, TX, USA).

## 4. Results

Among the n = 515 participants in this dataset, the median participant age at recruitment was 20 years (IQR: 18–22). Self-reported HIV prevalence was 3.7%, and nearly 80% of AGYW had resided in their community for more than five years (Table 1). Almost 97% of the participants had completed primary school education (i.e., Grade 9), and 77.3% were either attending high school (47.2%) or college/university (30.1%) before the COVID-19 lockdowns were imposed. Nearly 23% were not attending formal education when COVID-19-lockdowns were implemented, while 34% dropped out of enrolled education in 2020. Slightly over 10% did not use social support services from social workers within a year of the survey, 79.7% had access to family planning services, and 27.2% indicated that they were unable to obtain essential medication because of COVID-19 or the lockdown. The majority (73.8%) experienced HFI, almost 70% had household financial problems, while 22.5% went without a meal for a whole day at least once because of COVID-19. Thirteen percent experienced higher levels of violence at home during the lockdown.

**Table 1.** Exploration of participant characteristics (n = 515).

| Characteristic | Total N (%) | Characteristic | Total N (%) |
|---|---|---|---|
| Age at recruitment in years (median, IQR) | 20 (18–22) | | |
| Depressive symptoms score | | | |
| <12 | 359 (69.7) | | |
| ≥12 | 156 (30.3) | | |
| HIV status (self-reported) | | Lacked access to essential medication | |
| HIV positive | 17 (3.7) | Never | 367 (72.8) |
| HIV negative | 444 (96.3) | Often/Sometimes | 137 (27.2) |
| Length of stay in community | | Lacked access to family planning services | |
| <5 years | 104 (20.2) | Never | 401 (79.7) |
| ≥5 years | 410 (79.8) | Often/Sometimes | 102 (20.3) |
| Enrolled in fulltime study before lockdown | | Household financial problems | |
| Not enrolled | 117 (22.7) | Never | 154 (30.6) |
| High school | 243 (47.2) | Often/Sometimes | 349 (69.4) |
| College/University | 155 (30.1) | HFI (worry about food) (main explanatory variable) | |
| Disengaged in secondary or tertiary education in 2020 | | Never | 132 (26.2) |
| No | 481 (93.4) | Often/Sometimes | 371 (73.8) |
| Yes | 34 (6.6) | HFI (no meals for the whole day) | |
| Highest school grade completed | | Never | 390 (77.5) |
| <Grade 9 | 17 (3.3) | Often/Sometimes | 113 (22.5) |
| ≥Grade 9 | 498 (96.7) | Violence at home | |
| Lacked access to social support services | | Never | 438 (86.9) |
| Yes | 452 (89.7) | Often/Sometimes | 66 (13.1) |
| No | 52 (10.3) | Household asset index score * | 4.37 (3.5–5.6) |

* Based on Booysen, et al. [35].

Overall, 30.29% of the AGYW had depressive symptoms (Table 2). Comparing associations between demographic and socio-economic variables with depressive symptoms, significant differences between AGYW with and those without depressive symptoms were observed for the following variables: *age at recruitment* ($p = 0.016$), *enrolled in fulltime study before lockdown* ($p = 0.009$), *lacked access to essential medication* ($p = 0.005$), HFI ($p = 0.043$), *no meal for a whole day* ($p < 0.001$), and *violence at home* ($p = 0.003$) (Table 2). Almost three-quarters (73.76%) of the study population had HFI. One-third of the AGYW with HFI had depressive symptoms, which was higher than those without HFI ($p = 0.043$) (Table 3). Significant differences between AGYW with and without HFI were observed on median age at recruitment ($p = 0.021$), full study enrolment before lockdown ($p = 0.005$), financial problems ($p < 0.001$), and household asset index score ($p = 0.001$).

In regression analysis, the unadjusted model found that AGYW living in households with food insecurity were at an 87% higher risk of experiencing depressive symptoms [risk ratio (RR): 1.87; 95% CI: 1.47–2.38, $p < 0.001$)] compared to those from food-secure households (Table 4). In Model 1, adjusting for demographic characteristics increased the risk of depressive symptoms among AGYW in HFI by 1% ($aRR_1$: 1.88; 95% CI: 1.43–2.48, $p < 0.001$). Adjusting for *lacked access to social support services* in Model 2 increased the risk of AGYW living in food-insecure households experiencing depressive symptoms to 89% ($aRR_2$: 1.89; 95% CI: 1.43–2.51, $p < 0.001$). In Model 3, adjusting for *access to essential medicine* reduced the risk among AGYW from HFI experiencing depressive symptoms to 80% ($aRR_3$: 1.80; 95% CI: 1.35–2.42, $p < 0.001$). AGYW who lacked access to essential medication were 1.4 times (RR: 1.40: 95% CI: 1.20–1.64), $p < 0.001$) more likely to experience depressive symptoms than those who were able to access essential medication.

There was also a significant association between *age at recruitment* of AGYW and depressive symptoms, with younger women (based on median age [i.e., 20 (IQR:19–22)]) at 7% higher risk of experiencing depressive symptoms (RR:1.07; 95% CI: 1.01–1.14, $p < 0.020$). Sensitivity analysis (using *no meals for a whole day* as an alternate proxy for HFI) did not produce significant associations between HFI and depressive symptoms in univariable and multi-variable analyses.

**Table 2.** Comparison between AGYW with versus without depressive symptoms (n = 515).

| Characteristic | No Depressive Symptoms N (%) | Depressive Symptoms N (%) | *p*-Value | Characteristic | No Depressive Symptoms N (%) | Depressive Symptoms N (%) | *p*-Value |
|---|---|---|---|---|---|---|---|
| Total participants | 359 (69.71) | 156 (30.29) | | Lacked access to social-support services | | | |
| | **Median (IQR)** | **Median (IQR)** | | Yes | 321 (89.4) | 131 (90.3) | |
| Age at recruitment (years) | 20 (18–22) | 20 (19–22) | 0.016 | No | 38 (10.6) | 14 (9.7) | |
| | **N (%)** | **N (%)** | | Lacked access to essential medication | | | 0.005 |
| HIV status | | | 0.530 | Never | 274 (76.3) | 93 (64.1) | |
| HIV positive | 13 (4.0) | 4 (2.9) | | Often/Sometimes | 85 (23.7) | 52 (35.9) | |
| HIV negative | 308 (96.0) | 136 (97.1) | | Lacked access to family planning services | | | 0.095 |
| Length of stay in community | | | 0.180 | Never | 293 (81.6) | 108 (75.0) | |
| <5 years | 78 (21.8) | 26 (16.7) | | Often/Sometimes | 66 (18.4) | 36 (25.0) | |
| ≥5 years | 280 (78.2) | 130 (83.3) | | Household financial problems | | | <0.001 |
| Enrolled in fulltime study before lockdown | | | 0.009 | Never | 126 (35.2) | 28 (19.3) | |
| Not enrolled | 69 (19.2) | 48 (30.8) | | Often/Sometimes | 232 (64.8) | 117 (80.7) | |
| High school | 182 (50.7) | 61 (39.1) | | Household food insecurity (worry about food) (main explanatory variable) | | | 0.043 |
| College/University | 108 (30.1) | 47 (30.1) | | Never | 103 (28.8) | 29 (20.0) | |
| Disengaged in secondary or tertiary education in 2020 | | | 0.510 | Often/Sometimes | 255 (71.2) | 116 (80.0) | |
| No | 337 (93.9) | 144 (92.3) | | Household food insecurity (no meals for the whole day) | | | <0.001 |
| Yes | 22 (6.1) | 12 (7.7) | | Never | 296 (82.7) | 94 (64.8) | |
| Highest school grade completed | | | 0.650 | Often/Sometimes | 62 (17.3) | 51 (35.2) | |
| <Grade 9 | 11 (3.1) | 6 (3.8) | | Violence at home | | | 0.003 |
| ≥Grade 9 | 348 (96.9) | 150 (96.2) | | Never | 322 (89.7) | 116 (80.0) | |
| | | | 0.870 | Often/Sometimes | 37 (10.3) | 29 (20.0) | |
| | | | | Household asset index score | 4.37 (3.46–5.62) | 4.37 (3.86–5.62) | 0.93 |

**Table 3.** Comparison between AGYW with versus without household food insecurity (n = 515).

| Characteristic | Household Food Insecurity (Worry about Food), N (%) | | *p*-Value |
| --- | --- | --- | --- |
| | **Never** | **Often/Sometimes** | |
| Total participants | 132 (26.24) | 379 (73.76) | |
| | **Median (IQR)** | **Median (IQR)** | |
| Age at recruitment (years) | 19 (17–22) | 20 (18–22) | 0.021 [M] |
| | **N (%)** | **N (%)** | |
| HIV status | | | 0.860 [F] |
| HIV negative | 112 (96.6) | 328 (96.2) | |
| HIV positive | 4 (3.4) | 13 (3.8) | |
| Length of stay in community | | | 0.400 [C] |
| <5 years | 24 (18.2) | 80 (21.6) | |
| ≥5 years | 108 (81.8) | 290 (78.4) | |
| Enrolled in fulltime study before lockdown | | | 0.005 [C] |
| Not enrolled | 17 (12.9) | 98 (26.4) | |
| High school | 72 (54.5) | 161 (43.4) | |
| College/University | 43 (32.6) | 112 (30.2) | |
| Disengaged in secondary or tertiary education in 2020 | | | 0.170 [C] |
| No | 120 (90.9) | 350 (94.3) | |
| Yes | 12 (9.1) | 21 (5.7) | |
| Highest school grade completed | | | 0.910 [F] |
| <Grade 9 | 4 (3.0) | 12 (3.2) | |
| ≥Grade 9 | 128 (97.0) | 359 (96.8) | |
| Depression | | | 0.043 [C] |
| No | 103 (78.0) | 255 (68.7) | |
| Yes | 29 (22.0) | 116 (31.3) | |
| Financial problems | | | <0.001 [C] |
| No | 100 (75.8) | 54 (14.6) | |
| Yes | 32 (24.2) | 317 (85.4) | |
| Household asset index score | 5.11 (3.86–5.62) | 4.37 (3.41–5.62) | 0.001 [M] |

Note: [C]—Chi-Square test; [F]—Fisher's Exact; [M]—Mann–Whitney U.

**Table 4.** Regression modeling to identify associations between food insecurity and depressive symptoms with unadjusted risk ratio (RR) and adjusted risk ratio (aRR).

| Variable | Risk Ratio | Model 1 | Model 2 | Model 3 |
| --- | --- | --- | --- | --- |
| | **RR (95% CI) [*p*-Value]** | **aRR$_1$ (95% CI) [*p*-Value]** | **aRR$_2$ (95% CI) [*p*-Value]** | **aRR$_3$ (95% CI) [*p*-Value]** |
| Age at recruitment (years) | | 1.07 (1.01–1.14) [0.020] | 1.07 (1.01–1.14) [0.024] | 1.07 (1.01–1.14) [0.023] |
| Length of stay in community | | | | |
| <5 years | | ref | ref | ref |
| ≥5 years | | 1.24 (0.86–1.79) [0.255] | 1.23 (0.84–1.79) [0.294] | 1.20 (0.81–1.77) [0.363] |
| Highest school grade completed | | | | |
| <Grade 9 | | ref | ref | ref |
| ≥Grade 9 | | 0.83 (0.38–1.80) [0.641] | 0.83 (0.39–1.81) [0.650] | 0.82 (0.39–1.75) [0.616] |
| Household asset-index score | | 1.04 (0.98–1.10) [0.172] | 1.04 (0.98–1.10) [0.217] | 1.05 (0.99–1.10) [0.089] |

**Table 4.** *Cont.*

| Variable | Risk Ratio | Model 1 | Model 2 | Model 3 |
|---|---|---|---|---|
| | RR (95% CI) [p-Value] | aRR$_1$ (95% CI) [p-Value] | aRR$_2$ (95% CI) [p-Value] | aRR$_3$ (95% CI) [p-Value] |
| Lacked access to social support services | | | | |
| No | | | ref | ref |
| Yes | | | 0.90 (0.71–1.14) [0.391] | 0.86 (0.64–1.15) [0.320] |
| Lacked access to essential medication | | | | |
| Never | | | | ref |
| Often/Sometimes | | | | 1.40 (1.20 -1.64) [<0.001] |
| Household food insecurity (worry about food) | | | | |
| Never | ref | ref | ref | ref |
| Often/Sometimes | 1.87 (1.47–2.38) [<0.001] | 1.88 (1.43–2.48) [<0.001] | 1.89 (1.43–2.51) [<0.001] | 1.80 (1.35–2.42) [<0.001] |

## 5. Discussion

The COVID-19 pandemic amplified structural inequalities and HFI, increasing psychological distress among vulnerable populations. Globally, there have been calls to identify groups severely impacted by the pandemic to better direct recovery efforts toward high-risk groups and inform economic protection strategies during future pandemics [20,36]. AGYW in SSA are vulnerable to mental health problems, given the physiological, psychosocial, and cognitive changes experienced during this phase [10]. This vulnerable group has been neglected in COVID-19 recovery efforts [20]. The present study investigated the relationship between HFI and depressive symptomatology among AGYW during COVID-19, using cross-sectional survey data from South Africa. The vast majority of AGYW (73.8%) resided in food-insecure households. After adjusting for key socio-economic and demographic variables, AGYW living in households with food insecurity were at 80% (95% CI) higher risk of experiencing depressive symptoms compared to those from food-secure households. This highlights the need for stronger COVID-19 recovery investment towards multi-sectoral policies and programmes that promote the economic and psychological wellbeing of AGYW residing in food-insecure households.

In our sample, the prevalence of depressive symptoms during the COVID-19 pandemic was 30.3%. While few studies have reported depression outcomes among AGYW in SSA during COVID-19, our estimate is comparable with global estimates [18,37,38]. For example, the prevalence of depressive symptoms among AGYW in Kenya was estimated to be 34.5% [18], 32% in Peru [37], and 33% in the United States [38]. The Organization for Economic Cooperation and Development [39] (OECD) [35] reported a 31% prevalence of depression in the United Kingdom and 29% in both Japan and Belgium. The OECD also reported a global increase in depression among younger people during the COVID-19 pandemic, ranging from 30 to 80%.

Comparing the prevalence of depressive symptoms among AGYW from this cohort to previous South African studies proved challenging. This was in part due to the dearth of literature on the topic in SSA [40]. However, an earlier study among young South African people (aged 15–26 years) estimated the prevalence of depressive symptoms among young women at 20.5% [40]. A more recent estimate among 16–24-year-olds by Jesson, et al. [41] was much higher (48%). This high estimate is possibly due to the study's focus on a select group of young people residing in known high-risk areas of South Africa. Another study in the region examined the association between depressive symptoms and HIV incidence among AGYW (aged 13–21 years) and estimated a prevalence of depressive symptoms of 18.2% among this group [22]. Given the strong link between depressive symptoms and HIV acquisition, and HIV/AIDS being the leading cause of mortality and morbidity among AGYW in this region, policymakers should direct resources towards reducing depression among AGYW. Depression has also been linked to poor sexual reproductive health out-

comes (e.g., unplanned pregnancy and STIs). Programmes targeting depression could, therefore, assist in preventing unplanned pregnancies in AGYW. Healthcare policies should ensure that appropriate measures are in place to secure continued access to medication among vulnerable populations during times of disaster, as lack of access to needed medication was linked to an increased risk of depressive symptoms among AGYW during the COVID-19 lockdown.

Our study also found that AGYW living in households with food insecurity were at higher risk of depressive symptoms compared to those living in food-secure households (RR: 1.87; 95% CI: 1.47–2.38, $p < 0.001$) [26,27]. After adjusting for confounders, this risk decreased slightly (aRR$_3$, 1.80; 95% CI: 1.35–2.42, $p < 0.001$). This is also consistent with previous studies from SSA, which indicated that AGYW living in HFI were at higher risk of developing depressive symptoms in comparison to their male counterparts [17,18,42]. Gibbs, Govender and Jewkes [42], for instance, found that food-insecure AGYW from South African informal settlements were 5.57 times more likely to have depressive symptoms (aOR 5.57, $p = 0.039$).

Similarly, a study on Kenyan parent-adolescent dyads found that adolescents had 2.5 higher odds of depressive symptoms (OR 2.5, 95% CI: 2.0–3.1) [18]. Although several studies have highlighted the association between HFI and depression, it is worth noting that depression itself may be the reason for household food insecurity [43,44]. For example, Jesson et al. [41] found that younger people with probable depression had higher odds of being food-insecure (2.79, 95% CI: 1.57–4.94).

Previous studies in the region have found that HFI was exacerbated due to the COVID-19 pandemic [26,27]. In South Africa, formal and informal job losses during the COVID-19 pandemic may have directly contributed to HFI [45]. Job losses in this setting possibly meant that households lacked the income to purchase essential food items [46], which may have caused AGYW to become distressed [25] and develop depressive symptoms. HFI caused by job losses has been shown to be associated with caregiver anxiety, depression, and parenting stress [47]. In addition, poor HFI-induced parent/caregiver mental health is also associated with psychological distress in younger people [48]. Younger people also feel embarrassed by their household's food situation [48]. Feeling powerless, desperate, and guilty about their household's food situation could have directly contributed to their anxiety and depressive symptoms [49]. Such a state could lower adolescents' self-esteem and happiness and negatively affect their perceptions of their parental role model(s) [50]. If these socio-economic conditions persist without any coping strategies, this could have a longer-term negative effect on the health of young people. It is thus important to implement interventions that allow young people to easily access mental health services to prevent and/or manage depression.

### 5.1. Policy and Future Recommendations

The advent of COVID-19 and lockdowns worsened the economic situation of many households. Policies should prioritize identifying such households and provide them with relief (e.g., cash and food packages) to reduce HFI, thereby reducing the risk of depressive symptoms in these households. AGYW should also be given psychosocial support via building fortitude, defined by Pretorius and Padmanabhanunni [51] as a person's ability to manage stress and remain well. Fortitude is rooted in positive and/or adaptive appraisals of the self, family, and significant others.

Online and digital platforms could be offered to AGYW for mental health education and psychological counseling services, delivered via low-cost mobile phone applications [51,52]. These applications can also be used by trained community healthcare workers to disseminate knowledge about future pandemics, address uncertainties, and provide support [51]. Moreover, a strong focus should be placed on the early detection and management of depressive symptoms in AGYW. Family strengthening interventions focused on enhancing parenting, communication, and social connectedness should be prioritized, with an emphasis on building resilience among AGYW [53]. There is, therefore, a need for more

intersectionality research to look at findings by geography, ethnicity, and demography to ensure nuanced policy decision-making [54]. In addition, scaling up food programmes during future pandemics could be effective in protecting the mental health of vulnerable AGYW in South Africa.

### 5.2. Study Strengths

The strengths of the study include the following: (1) the survey was conducted in multiple South African districts; (2) the study used a robust depressive-symptoms measure (CES-D-10) previously validated in South Africa [31]; and (3) data collectors conducted interviews in participants' preferred languages and built a good rapport with AGYW to make them feel safe over the phone.

### 5.3. Limitations

This study had the following limitations: (1) the main explanatory variable, HFI, was based on one item, namely, *worry about food running out because of a lack of money*. To account for this shortcoming, a sensitivity analysis using *no meals for the whole day* as a proxy for HFI was performed and found no significant associations with depressive symptoms. Future studies should be designed using validated measures, such as the Food Insecurity Experience Scale, to measure HFI in population surveys [55]. (2) This analysis was based on a cross-sectional dataset. Therefore, causation cannot be established. It is possible that household violence or substance misuse may have been confounding the relationship, as families experiencing higher levels of violence or exposed to substance misuse may experience greater levels of HFI [56], and thus, AGYW residing in these homes could have higher levels of depressive symptoms. Furthermore, it is plausible that a coincident variable, such as district-level economic development, may also be influencing the relationship. Households in areas with lower economic development may have lower access to job opportunities, thus increasing HFI levels and, subsequently, depressive symptoms among adolescents in these homes. There is thus a need for a longitudinal study to examine the temporality of HFI-depression causation [57]. (3) The study sample was not a national sample; as such, the results might not be generalizable over the entire AGYW population of South Africa. (4) Selection bias, the AGYW in this study were selected specifically because they were enrolled in a programme for AGYW (the My Journey programme). (5) The parent study did not collect data on all key factors that can impact mental health, such as social media use. (6) The study was performed during the COVID-19 pandemic. We did not sample this population pre-COVID-19. Hence, we were unable to perform pre-post COVID-19 analyses of study findings.

## 6. Conclusions

This study found that during the COVID-19 pandemic, AGYW who experienced HFI were at higher risk of exhibiting depressive symptoms compared to those living in food-secure households. This study sheds light on the effect of HFI on the mental health of AGYW living in these households during the COVID-19 pandemic. As part of COVID-19 recovery efforts, there is a need to scale up intensive AGYW mental-health screening programmes, given the burden of depressive symptoms in this population. Our findings suggest that using HFI would be an effective indicator in community-based screening programmes for identifying AGYW at risk of depressive symptoms and linking them to mental health services. Our results highlight that multi-sectorial approaches and programmes are critical for reducing the mental-health burden among AGYW. Social policies should include food hampers, cash transfers, support groups, and social-protection interventions that target younger people. Economic policies should also focus on mobilizing youth economic development by investing in entrepreneurship development.

**Author Contributions:** S.C. and D.G. prepared and wrote the manuscript. C.M. and C.W. critically reviewed and edited the manuscript. D.G. formulated the ideas and overarching goals and aims and led the compilation of the manuscript. L.N.S. and A.M. conducted all statistical analyses and data syntheses. All authors have read and agreed to the published version of the manuscript.

**Funding:** This research was funded by the Networking HIV and AIDS Community of Southern Africa (NACOSA). The AGYW intervention was funded by the Global Fund to Fight AIDS, TB and Malaria. The combination HIV prevention interventions were implemented in 12 districts in South Africa by a range of civil society organisations that were appointed by the organisations responsible for the management of the AGYW programme: NACOSA; the AIDS Foundation of South Africa (AFSA) and Beyond Zero. The programme was aligned with the She Conquers campaign and was implemented with support from the South African National AIDS Council (SANAC) through the Country Coordinating Mechanism (CCM) and the CCM Secretariat.

**Institutional Review Board Statement:** The study was conducted in accordance with the Declaration of Helsinki and approved by the Ethics Committee of the South African Medical Research Council (EC036-9/2020).

**Informed Consent Statement:** Written informed consent was obtained from the participants involved in the study to publish this paper. Parental/caregiver consent was obtained for participants younger than 18 years old prior to conducting the assent procedures.

**Data Availability Statement:** The raw data were generated by the South African Medical Research Council. The derived data supporting the findings of this study are available from the corresponding author [SC] on request.

**Acknowledgments:** We would like to thank the AGYW who participated in this research and shared their views and experiences with us. We acknowledge the implementers of the AGYW programme for their support in the evaluation study and for providing support and counselling to the participants referred to them. We also thank our excellent team of data collectors and monitors, and individuals who provided administrative and logistical support for the study.

**Conflicts of Interest:** The authors declare no conflicts of interest.

## Appendix A

**Table A1.** Description of variables.

| Variables | Survey Questions | Survey Response Options | Grouping and Recoding of Survey Response Options |
|---|---|---|---|
| Age at recruitment | When were you born? If you are not sure, please estimate your date of birth | dd-MM-yyyy [Min 01-01-1996, Max 23-11-2005] | N/A |
| Length of stay in community | How long have you been living in this community? | 1 = Less than a year<br>2 = About 1 year<br>3 = About 2 years<br>4 = About 3 years<br>5 = About 4 years<br>6 = About 5 years<br>7 = More than 5 years<br>99 = Prefer not to answer | 1–5 = [<5 years]<br>6–7 = [≥5 years] |
| Enrolled in fulltime study before lockdown | In 2020, before COVID-19 and the lockdown, were you enrolled in school, or college or university fulltime? | 1 = I was enrolled in high school<br>2 = I was enrolled in a TVET college<br>3 = I was enrolled in another type of college<br>4 = I was enrolled in university<br>5 = I was enrolled in another training institution<br>6 = I was not enrolled in any educational institution<br>99 = Prefer not to answer | 1 = [High school]<br>2–5 = [College/University]<br>6 = [Not enrolled] |
| Disengaged in secondary or tertiary education in 2020 | During 2020, did you drop out of any school, college, university or other educational institution? | 1 = Yes<br>2 = No<br>99 = Prefer not to answer | 1 = [Yes]<br>2 = [No] |

**Table A1.** *Cont.*

| Variables | Survey Questions | Survey Response Options | Grouping and Recoding of Survey Response Options |
|---|---|---|---|
| Highest school grade completed * | What is the highest grade of school you have completed? | 0 = Not completed any grade<br>1 = Grade 1<br>2 = Grade 2<br>3 = Grade 3<br>4 = Grade 4<br>5 = Grade 5<br>6 = Grade 6<br>7 = Grade 7<br>8 = Grade 8<br>9 = Grade 9<br>10 = Grade 10<br>11 = Grade 11<br>12 = Grade 12<br>99 = Prefer not to answer | 1–8 = [<grade 9]:<br>9–12 = [≥grade 9]: |
| Lacked access to social support services ** | What kind of problems has the social worker or counsellor helped with in the past year? Choose as many answers as relevant. | 1 = Getting an ID or birth certificate<br>2 = Getting a child support grant<br>3 = Helped me when I was feeling sad or depressed<br>4 = Helped me with problems at school<br>5 = Helped me with the abuse I've experienced<br>6 = Helped me with rape<br>7 = Other help<br>99 = Prefer not to answer | 1 = [Yes to any of 1–7]<br>0 = [None of any of 1–7 selected] |
| Lacked access to essential medication | You were unable to get medicine you need because of COVID-19 or the lockdown | 1 = Never<br>2 = Sometimes<br>3 = Often<br>4 = I did not need medicine<br>99 = Prefer not to answer | 1 and 4 = [Never]<br>2–3 = [Often/Sometimes] |
| Lacked access to family planning services | You were unable to get family planning (birth- control pills, injection, implant) you needed because of COVID-19 or the lockdown. | 1 = Never<br>2 = Sometimes<br>3 = Often<br>4 = I did not need family planning<br>99 = Prefer not to answer | 1 and 4 = [Never]<br>2–3 = [Often/Sometimes] |
| Household financial problems | During the lockdown, me or my family experienced financial (money) problems | 1 = Never<br>2 = Sometimes<br>3 = Often<br>99 = Prefer not to answer | 1 = [Never]<br>2–3 = [Often/Sometimes] |
| Household food insecurity (worry about food) | During the lockdown I worried that my food would run out because of a lack of money | 1 = Never<br>2 = Sometimes<br>3 = Often<br>99 = Prefer not to answer | 1 = [Never]<br>2–3 = [Often/Sometimes] |
| Household food insecurity (no meals for the whole day) | During the lockdown did you go a day and night without eating because of lack of food? | 1 = Never<br>2 = Sometimes<br>3 = Often<br>99 = Prefer not to answer | 1 = [Never]<br>2–3 = [Often/Sometimes] |
| Violence at home | During the lockdown, was there more violence in your home? | 1 = Never<br>2 = Sometimes<br>3 = Often<br>99 = Prefer not to answer | 1 = [Never]<br>2–3 = [Often/Sometimes] |
| Household asset index score ***<br>[rw+tw+fw+toiw+ww] | rw+tw+fw:<br>At home, do you have any of these things in working condition? | Fridge<br>TV<br>Radio | Fridge (fw):<br>1 = yes; 0 = no<br>fw = 1.682 if fridge = 1<br>fw = −0.096 if fridge = 0<br>TV (tw):<br>1 = yes; 0 = no<br>tw = 1.608 if tv = 1<br>tw = −0.1 if tv = 0<br>Radio (rw):<br>1 = yes; 0 = no<br>rw = 0.282 if radio = 1<br>rw = −0.225 if radio = 0 |

**Table A1.** *Cont.*

| Variables | Survey Questions | Survey Response Options | Grouping and Recoding of Survey Response Options |
|---|---|---|---|
| toiw: What type of toilet do you have? | 1 = Own flush toilet<br>2 = Shared flush toilet with other household/s<br>3 = Bucket latrine<br>4 = Pit latrine<br>5 = Ventilated pit latrine<br>6 = Other<br>99 = Prefer not to answer | toiw = 1.164 if toilet = 1–2<br>toiw = −0.088 if toilet = 3–5<br>vtoiw = −0.16 if toilet = 6 |
| ww: At home, where do you get your drinking water from? | 1 = Tap in house<br>2 = Tap in yard<br>3 = Tap in community<br>4 = Well or borehole<br>5 = River or stream<br>6 = Rainwater tank<br>7 = Water truck<br>8 = Other<br>99 = Prefer not to answer | ww = 0.885 if water = 1–2<br>ww = −0.026 if water = 3<br>ww = −0.225 if water = 4<br>ww = −0.222 if water = 5<br>ww = −0.197 if water = 6–8 |

\* Grade 9 is the minimal or compulsory level of education; \*\* a "yes" response indicates that at least one form of support/service was received from a social worker (e.g., obtaining ID or birth certificate, receiving child support, help when feeling sad or depressed, help with problems at school, help with abuse, or help with rape); \*\*\* is an index score used to gauge poverty. It considers whether a household has a radio, television, fridge, a toilet, and water, as per Booysen, van der Berg, Burger, Maltitz and Rand [35].

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
