# Peer review of "Associations between Household Food Insecurity and Depressive Symptomology among Adolescent Girls and Young Women during the COVID-19 Pandemic in South Africa"

_adolescents, doi:10.3390/adolescents4010013_

Round 1

Reviewer 1 Report

Comments and Suggestions for Authors

This study appears to build upon previous work in this area.   

The article indicates that this is a secondary data review, however, then later indicates the participates were those able to be contacted by phone leading reviewer to reread to determine if this was a primary or secondary data study.  Clarification is needed.

This study demonstrates the impact of the COVID-19 pandemic on mental health status among women in food insecure households

Comments on the Quality of English Language

Minor editing is suggested.   I believe it would provide clarification throughout the article

Reviewer 2 Report

Comments and Suggestions for Authors

The authors did a good job of presenting their study and the findings. First person is used in parts of the article, which I recommend changing to third person. Where you able to control for the confounding variable of hours of social media usage and negative mental health impact? If so, mention in discussion. If not, add to limitation. FYI: there are so articles in the scientific literature on this topic to pull for reference.

Line 74 add source number for website.

Lines 186: Delete “high” given the discussion that follows, it was essentially the same or a bit lower than found in other studies. Suggest change to “…depressive symptoms during the COVID-19 pandemic was 30.3%.”

Line 223: Is this statement accurate? “may be the cause of food insecurity” or is depression associated with food insecurity, or is depression associated with a higher risk of food insecurity?

Line 234: “cause psychological distress in younger people” Double check to make sure the study reported a causal relationship vs. an association.

Comments on the Quality of English Language

Minor edits only. First person is used in parts of the article, which I recommend changing to third person.

Reviewer 3 Report

Comments and Suggestions for Authors

His proposal is adequate and well argued, although the current context given that the pandemic has ended is obsolete, perhaps in a specific monograph or COVID concretion scenario it will have an impact. Continue in your line of work, as I indicated, it is of quality but out of context.

Comments on the Quality of English Language

Not necessary

Reviewer 4 Report

Comments and Suggestions for Authors

The paper is of an applied nature and broadens the understanding of the pandemic’s effect on mental health of people.

Questions and remarks:

1. For a full appreciation of the manuscript and the intent of the study, referring the study through references seems inappropriate. The reader should have a general clear picture of the study without seeking for additional information about the study and the authors’ intent.

2. The title of the article indicates ‘mental health’, in the study the scale of depression was evaluated, which is a significantly narrower concept than mental health.

3. The analysis of the results is not sufficient and is often replaced by other studies.

4. It was not clear which of the variables had been allocated to the food security domain.

5. Annex 2 is a duplicate of Table 3.

6. How can the results of this study be used to prove that the results are related to the epidemic of COVID-19? It is possible that these are characteristics that are common to all adolescent girls and young women in South Africa.

7. The result of the study is an obvious one, and it lies on the surface. Why is there a need for secondary data analysis when it has been repeatedly demonstrated that food insecurity affects people's mental health? And the same evidence is cited in the discussion section. So the question is: what is the novelty of the study and of this article?

8. The Microsoft Word template or LaTeX template was not used for the submission of the article to Adolescents journal.

Reviewer 5 Report

Comments and Suggestions for Authors

The association between household food insecurity and mental health among adolescent girls and young women (AGYW) during the COVID-19 epidemic in South Africa

This is interesting study, however, the authors should be more requirement for several suggestion to modify the manuscript as follows:

Format

Should be strictly followed the MDPI format is required. I found format, writing styles, and interpretative introduction, methods, findings, discussion, and conclusion are not logical academic writing.

Title

The title is not precise and invalid meaning, theoretical reflection, and context of study. The precise title should be "The impacts of Household Food Insecurity on Mental Health among Adolescent during the COVID-19 Epidemic"

Abstract

It is not clear what the main purpose of the study, utilizing methods, the key findings, conclusion, and practical implications. See abstract section.

Introduction

Should be cleared what phenomena is, especially among adolescents in South Africa. See the first paragraph (line 1-16)

Should be clearly of main problems of mental health among adolescents in South Africa (see line 18-22). Then explained based on phenomena links to theoretical gaps is required.

What theoretical existing (see line 24-34)

What/how/why the authors needs to explore/examine/investigate? How many factors/issues related to mental health among adolescents?

Lastly, provided a clear purpose of study is required (line 41-50).

Note: Should be thought what the main problems of mental health among adolescents during the COVID-19 epidemic in South Africa. What key concepts are you providing to operationalize in the study? What gaps of knowledge you try yo exist? What objectives/questions are your needs to explore?

Should be clearly how many factors/issues are included in the testing.

Methods

The authors have made a concerted effort to address design, data, measurement, and analysis. This is why I found, the author jumped into measure (outcomes), my question is, how and what methods are you applying for?

1. Should be more detailed what the study design is.

2. What sample/sample applied in the study?

3. How measures are? How many items are included? What factors do you operationalize based on who?

Potential correlates are assessed items or intercorrelation among variables, what? I think, it is not potential correlated. Should be clear what?

Results

Where is descriptive of samples?

Should be more explained what the Model 1 through Model 3 are correlated/significant. What each model mean? What values are communicated for? What positive and negative reflection to the findings?

Discussion

Should be separated discussion section and conclusion section.

1. Should be cleared what are the main findings.

2. Should be cleared what are the key results supported or contrast other studies.

3. Should be included (1) practical implications, (2) theoretical contributions, (3) limitations and future direction in the discussion section is required.

Conclusion

Should be concluded with the main findings is required.

Refs

Should be strictly followed MDPI format is required. I found many refs are not updated and not relevant to the study. Should be revised all refs are required.

Comments on the Quality of English Language

I think it should be required major corrections of the English language, so authors should check the manuscript in detail. Many grammatical errors, invalid meaning, and poor academic communication. Provided native speaker proofread/edit required.

Round 2

Reviewer 2 Report

Comments and Suggestions for Authors

The authors did a good job addressing my concerns. I think the paper is ready for publication.

Author Response

We thank the reviewer for supporting the publication of our work..

Reviewer 3 Report

Comments and Suggestions for Authors

The proposal, despite being interesting at a scientific level, presents serious methodological errors in selection/sampling which reduce its scientific rigor. Keep them in mind for future proposals.

Comments on the Quality of English Language

They are not serious errors, there is no need to review them.

Author Response

We thank the reviewer for their comment. Unfortunately, this study was based on secondary data analysis. As such, data selection/sampling was beyond the authors’ control. However, valuable lessons have been learnt from this secondary data analysis in regard to the point raised by the reviewer. The researchers will definitely be mindful of the valuable insights shared and will apply recommendations to future research.

Reviewer 4 Report

Comments and Suggestions for Authors

The authors have taken the comments into acount and revised the manuscipt, It has become clearer and more structured. However, the title of the article is quite long and includes an abbreviation in parentheses (AGYW). It is up to the editor to decide whether this is acceptable.

Author Response

We thank the reviewer for their comment. We have now removed (AGYW) from the title as there is no pressing need to have it included. We hope that this is acceptable and, to some extent, contributes towards reducing the length of the title.

Reviewer 5 Report

Comments and Suggestions for Authors

All comments are revised and well-suited for publication in the Adolescents.

Comments on the Quality of English Language

Minor errors and meaning.

Author Response

We thank the reviewer for their support in our work being published.